# 3D Printed Gene-Activated Sodium Alginate Hydrogel Scaffolds

**DOI:** 10.3390/gels8070421

**Published:** 2022-07-06

**Authors:** Maria A. Khvorostina, Anton V. Mironov, Irina A. Nedorubova, Tatiana B. Bukharova, Andrey V. Vasilyev, Dmitry V. Goldshtein, Vladimir S. Komlev, Vladimir K. Popov

**Affiliations:** 1Institute of Photon Technologies of Federal Scientific Research Centre “Crystallography and Photonics”, Russian Academy of Sciences, Moscow 108840, Russia; khvorostina.m@gmail.com (M.A.K.); scftlab@gmail.com (A.V.M.); 2Research Centre for Medical Genetics, Moscow 115478, Russia; irina0140@gmail.com (I.A.N.); bukharova-rmt@yandex.ru (T.B.B.); vav-stom@yandex.ru (A.V.V.); dvgoldshtein@gmail.com (D.V.G.); 3Central Research Institute of Dental and Maxillofacial Surgery, Moscow 119021, Russia; 4A.A. Baikov Institute of Metallurgy and Materials Science, Russian Academy of Sciences, Moscow 119334, Russia

**Keywords:** gene-activated scaffolds, 3D printing, sodium alginate hydrogel, plasmid DNA

## Abstract

Gene therapy is one of the most promising approaches in regenerative medicine to restore damaged tissues of various types. However, the ability to control the dose of bioactive molecules in the injection site can be challenging. The combination of genetic constructs, bioresorbable material, and the 3D printing technique can help to overcome these difficulties and not only serve as a microenvironment for cell infiltration but also provide localized gene release in a more sustainable way to induce effective cell differentiation. Herein, the cell transfection with plasmid DNA directly incorporated into sodium alginate prior to 3D printing was investigated both in vitro and in vivo. The 3D cryoprinting ensures pDNA structure integrity and safety. 3D printed gene-activated scaffolds (GAS) mediated HEK293 transfection in vitro and effective synthesis of model EGFP protein in vivo, thereby allowing the implementation of the developed GAS in future tissue engineering applications.

## 1. Introduction

Various 3D printing technologies are currently being used to fabricate bioactive scaffolds of a given architectonics with certain biochemical characteristics for a wide range of tissue engineering and drug delivery applications [1,2,3,4,5,6]. One of the most promising directions in this field is the development of three-dimensional structures that provide localized gene, e.g., DNA, and RNA, delivery to injured tissues and organs for more effective stimulation and support of regenerative processes due to the induced target genes expression in host cells [7,8,9].

The efficiency and safety of gene delivery into living cells is the key requirement for the successful development of bioactive scaffolds based on gene-activated materials [10]. Among the existing variety of different approaches, the non-viral gene delivery systems, involving plasmid DNA and polycationic polymers as transfecting agents, should be particularly highlighted [11]. Firstly, it is due to the well-known polycation’s ability to bind pDNA into stable and condensed polyplexes [12]. Moreover, these complexes protect pDNA from enzymatic degradation in vivo and at the same time reduce their negative charge, thereby ensuring effective and safe target cell transfection [13,14].

Scaffolds based on various biocompatible and bioresorbable materials with specified physicochemical and biological properties are usually applied to support localized and sustained gene delivery [15]. They provide a microenvironment for infiltration, adhesion, and proliferation of certain cell types, at the same time allowing the embedded genetic constructs to direct cellular differentiation and induce regeneration of damaged tissues [16]. However, despite many studies and real achievements in this field, the search and selection of optimal materials, as well as fabrication methods, to form highly effective gene-activated scaffolds (GAS) are still relevant and challenging in biomaterial science and tissue engineering [17]. 

Sodium alginate (SA) is one of the natural polymers commonly used in tissue engineering, primarily due to its high biocompatibility and no immunogenicity, as well as the simplicity of fabricating various hydrogels based on it [18]. The gentle sodium alginate polymerization process with divalent ions, e.g., Ca^2+^, Ba^2+^ etc., proceeding under normal conditions and eliminating the use of toxic components, makes it easy to combine this biopolymer with biologically active molecules, plasmid DNA in particular, without loss of its properties and functionality, which is crucial for regenerative medicine applications [19,20].

In recent years, various 3D printing techniques have attracted much attention as perspective methods to produce three-dimensional scaffolds based on SA [21,22]. The layer-by-layer plotting of the initial material according to the computer-aided model (CAM) and its subsequent gelation followed by cross-linking enable rapidly forming structures of almost any complexity. 3D printed scaffolds mimic the architectonics of the restored tissue fragment ensuring the ultimate integration with surrounding tissues and migration of progenitor cells deep into scaffold volume with its subsequent vascularization and innervation [23]. Moreover, the ability to control both the amount embedded in a scaffold DNA and the kinetics of its release in vivo, ensures the maintenance of the required therapeutic dose in the desired location [24].

This study aimed to develop a universal (in terms of using various genetic materials) platform for the gene-activated SA-based scaffolds fabrication using the original 3D cryoprinting methodology [25]. The possibility of incorporating polyplexes with model plasmid DNA, encoding the enhanced green fluorescent protein (pEGFP), into 3D printed hydrogel scaffolds with preservation of their transfecting ability both in vitro and in vivo has been demonstrated for the first time.

## 2. Results 

### 2.1. Transfection Efficiency of pEGFP with TF and PEI

Initially, to assess the degree of pEGFP condensation in polyplexes containing TF or PEI, the hydrodynamic diameters and zeta-potentials of plasmid DNA, transfecting agents, and polyplexes were measured using dynamic light scattering (DLS) (Table 1). pEGFP with the average hydrodynamic diameter of 420 ± 70 nm was shown to be condensed by transfecting agents by three and five times for TF and PEI, respectively. Zeta-potential of pEGFP/PEI polyplexes was much higher than that of pEGFP/TF polyplexes, suggesting the PEI potential to promote more efficient pDNA transfer through the cell membrane. 

When evaluating the polyplexes’ transfecting ability during their incubation with HEK293 (Figure 1), 13 ± 3% and 30 ± 6% of seeded cells were transfected with pEGFP/TF and pEGFP/PEI polyplexes, respectively. Thus, polyethyleneimine as a non-viral vector was shown to exhibit a more effective transfecting ability compared to TurboFect, which makes it more preferable for the plasmid DNA delivery into cells.

### 2.2. In Vitro Biocompatibility Assessment

MTT tests (Figure 2a) showed that there was a statistically significant (*p* < 0.05) change in the relative cell viability after seven days of incubation in the presence of the non-cross-linked sodium alginate affecting MTT test results for 3D printed SA-based scaffolds. However, the increase of crosslinking agent concentration and addition of genetic constructions did not provoke cell death and relative cell viability was still above 70% which indicated the lack of cytotoxic effects in MSCs. 3D printed scaffolds were shown to promote cell adhesion (Figure 2b,c) allowing MSCs to occupy and attach to the scaffold surface. Thus, 3D printed SA scaffolds were demonstrated to possess biocompatible properties and be suitable for GAS fabrication.

### 2.3. Bioresorption of Gene-Activated Structures

The mesh-like structures of SA scaffolds crosslinked with 2 wt.% and 10 wt.% CC aqueous solutions during their incubation with MSCs are shown in Figure 3a. Figure 3b demonstrates a change in the swelling degree of SA fibers with time. It should be noted that alginate scaffolds crosslinked with a 2 wt.% CC solution were almost completely resorbed after seven days of their incubation in the presence of MSCs, while their analogs, crosslinked with a 10 wt.% CC solution maintained structural stability for 21 days. At the same time, the average diameter of their fibers increased by an average of 51%.

### 2.4. The Genetic Constructs Stability in GAS

The influence of the 3D cryoprinting process on the structural integrity of plasmid DNA is presented in Figure 4. Incorporated in SA scaffold pEGFP released into saline solution was detected after seven days with a concentration of 750 ng/mL close to the lower sensitivity threshold of the spectrophotometer (Figure 4a). 

The peak of pEGFP absorbance at a wavelength of 257 nm corresponding to initial double-stranded plasmid DNA (Initial pEGFP) was preserved during 3D printing. The ability of plasmid DNA to penetrate the cell and promote the target protein synthesis was demonstrated using HEK293 cell cultures when incubating with released pEGFP and PEI for 24 h (Figure 4b–e). 

### 2.5. In Vivo Biocompatibility Assessment

During histological dehydration, the scaffold material was separated from the tissues forming empty white spaces in some cases (Figure 5). The mesh-like scaffolds were surrounded by granulation tissue — connective tissue rich in full-blooded vessels and young fibroblasts—infiltrated by segmented leukocytes and lymphocytes (Figure 5g). Their total number on histological sections was 110 ± 20 cells/mm^2^ for control samples, 750 ± 90 cells/mm^2^ for samples with the plasmid, and 1000 ± 100 cells/mm^2^ for samples with polyplexes. An accumulation of macrophages was detected near the scaffold material, which indicates active resorption of hydrogel scaffolds.

### 2.6. Transfection Efficiency In Vivo

Sections with specific immunohistochemical detection of EGFP are shown in Figure 6. This indicates that implanted 3D-printed gene-activated SA scaffold mediates pEGFP transfection in vivo with the subsequent synthesis of EGFP in tissue cells. SA scaffolds embedded with polyplexes are shown to promote more effective transfection in comparison with their analogs containing only plasmids (Figure 6b,c). 

The distance from the scaffold surface at which EGFP synthesizing cells were still observed was 20 ± 5 µm for scaffolds containing pEGFP and 90 ± 20 µm for scaffolds containing pEGFP/PEI polyplexes (Figure 7a). Pure SA did not mediate any transfection. The number of transfected cells in the SA scaffold implantation zone is shown in Figure 7b.

## 3. Discussion

The plasmid DNA encoding EGFP gene was chosen as a model pDNA ensuring a straightway visualization of the transfection region by transgene expression thereby significantly simplifying the in vitro transfection verification. The EGFP gene expression was studied using the HEK293 cell line, which can be easily transfected with most transfecting agents and guarantees proper protein folding and its posttranslational modifications [26,27]. Therefore, the transfecting ability of polycationic polymers became possible to evaluate. Polycations can condense nucleic acids to mediate and facilitate the penetration of negatively charged molecules into the cell, indicating that the transfection efficiency depends directly on the polyplexes’ size and zeta-potential [28]. Polycationic vectors implemented by different research groups were well-proven for use with most of the cell cultures [29,30]. Based on the comparative experiment results, PEI was selected in this study as the most effective transfection agent, which showed a more noticeable plasmid condensation degree and positive charge compared to TurboFect, as well as a stronger transfecting activity.

Sodium alginate-based hydrogels with embedded genetic constructions are known to hold the potential to provide localized and sustained gene delivery. For example, an injectable hydrogel containing a plasmid with the BMP-2 gene was developed in [31] to restore bone tissue. However, the demonstrated polymerization method did not allow the formation of specified porous structures. Lee’s group attempted to control the shape and thereby the dose of biologically active molecules during the formation of alginate microparticles using vibrational nozzle technology [32]. Microparticles exhibited excellent osteoinductive properties but still could be applied only to a limited amount of bone defect models.

The development of additive manufacturing (3D printing) technologies offered an opportunity to reproduce complex three-dimensional structures with the desired precision and efficiency, thereby allowing another way to control the release kinetics of genetic constructs and, at the same time, improving the GAS integration with tissues in the implantation site. K. Sun’s group introduced a multi-nozzle system for the layer-by-layer formation of SA 3D structures [22]. W. Zhou’s group implemented preliminary sodium alginate polymerization using D-gluconic acid d-lactone to improve the rheological properties of its aqueous solution [33]. Original 3D cryoprinting technique with Ca^2+^ ions as crosslinking agents, implemented in our study provides a means to fabricate porous three-dimensional scaffolds of a given architectonics without any additional experimental setup complications or preparatory procedures, meeting the requirement for structural safety and integrity of the genetic material embedded in GAS.

The physicochemical and biological properties of alginate scaffolds, which play a crucial role in the successful development of gene-activated matrices, may vary depending on the concentration of the crosslinking agent used. Calcium ions possessed biocompatible properties having no cytotoxic effect in vitro. An increase in its concentration during the crosslinking process slowed the swelling and, consequently, biodegradation of the scaffold structures [34], as well as to preserve their stability for three weeks to maintain the necessary microenvironment for cell proliferation and differentiation in vivo [35,36]. 

The insignificant inflammatory reaction caused by gene-activated SA-based scaffolds during their intramuscular implantation, in our opinion, is due to the moderate toxic effect of the incorporated transfecting complexes when they are released into the surrounding tissues, as well as an increase in the local divalent ion concentration leading to the attraction of leukocytes from blood vessels to the implantation zone [37]. However, these results, in general, correspond to the standard foreign body reaction to gradually degrading biomaterials [38]. 

It was shown that transfection in vivo is characterized by the distribution of transfected cells near the GAS surface and a predictable increase in the depth and area of transfection when using PEI as a transfecting agent. The promising results of the IHC analysis confirmed the prospect of implementing the developed 3D scaffolds for target gene delivery to damaged tissues to facilitate regenerative processes. The results of this study have presented the fundamental possibility to form the universal (in terms of using various genetic materials) platform for the gene-activated sodium alginate hydrogel scaffolds fabrication using the original 3D cryoprinting methodology. Our experimental in vitro and in vivo assessments have demonstrated the ability of such structures not only to promote the target protein expression but also to be promising constituents of tissue-engineered constructs for effective gene delivery.

## 4. Conclusions

The results of this study have presented the fundamental possibility to form the universal (in terms of using various genetic materials) platform for the gene-activated sodium alginate hydrogel scaffolds fabrication using the original 3D cryoprinting methodology. Our experimental in vitro and in vivo assessments have demonstrated the ability of such structures not only to promote the target protein expression but also to be promising constituents of tissue-engineered constructs for effective gene delivery.

## 5. Materials and Methods

### 5.1. Plasmid DNA

Plasmid DNA encoding enhanced green fluorescent protein (pEGFP, Clontech, Mountain View, USA) was used as a model genetic construct. Plasmids were amplified in Escherichia coli in LB Broth medium (Merck KGaA, Darmstadt, Germany) with 50 µg/mL of kanamycin (GRISP, Porto, Portugal) and isolated with a Zymo Research Plasmid Midiprep Kit (Zymo Research, Irvine, CA, USA) according to the manufacturer’s protocol.

### 5.2. Plasmid DNA Delivery

TurboFect (TF, Thermo Fisher Scientific, Waltham, MA, USA) and polyethyleneimine (PEI, linear, 25 kDa, Polysciences, Warrington, PA, USA) were used as transfecting agents (TA) to mediate the plasmid DNA delivery into cells.

The plasmid DNA condensation degree using transfecting agents was determined by dynamic light scattering (DLS) using Zetasizer Nano ZS (Malvern Instruments Ltd., Malvern, UK). 1 µg of pEGFP, 2 µg of TF, and 3 µg of PEI were separately dissolved in 1 mL of deionized water (diH2O, pH 7.0). Polyplexes were prepared according to the manufacturers’ recommendations: 1 µg pEGFP: 2 µL TF and 1 µg pEGFP: 3 µg PEI, respectively, were kept in 100 µL of diH2O at 37 °C for 30 min and then diluted with 900 µL of water to prevent multiple scattering. Measurements were carried out at 37 °C with a light scattering angle of 90°.

### 5.3. 3D Cryoprinting 

8 g of sodium alginate (Chimmed, Russia) was dissolved in 92 mL of water to obtain 8 wt.% solution, which was subsequently sterilized by UV radiation for 20 min. 100 µg of pDNA or polyplexes with 100 µg of pDNA were separately suspended in 100 mg of hydrogel to obtain gene-activated materials. Aqueous solutions of calcium chloride (CC, Merck KGaA, Darmstadt, Germany) with concentrations from 2 wt.% to 10 wt.% were used as a source of Ca^2+^ ions for intermolecular crosslinking and gelation of sodium alginate.

The GAS fabrication was performed using our previously developed method of three-dimensional cryoprinting [25] and a custom-designed 3D printer, presented in Figure 8.

3D cryoprinting is based on the layer-by-layer extrusion plotting of viscous materials (“ink”) on the selected substrate according to the coordinates specified by the original CAM under a significant (30–60 °C) negative temperature gradient between the printer dispenser nozzle and the substrate. During this process, a strictly dosed amount of material undergoes a rapid thermally induced phase transition or a change in its aggregate state resulting in the formation of solid, “frozen” elements of a reproducible three-dimensional object. A motorized dispenser providing a working pressure of 0.1 ÷ 0.4 MPa and a set of replaceable stainless-steel nozzles with an internal diameter from 100 to 1000 µm are used to dose materials of various viscosities from 0.01 to 10 Pa × s. Peltier elements operating in both heating and cooling modes allow to change and maintain the dispenser and the substrate temperatures in the ranges from 15 to 45 °C and from −30 to 0 °C, respectively. The set temperature difference is maintained with an accuracy of up to 1 °C by the automated control system of the 3D printer.

This approach enables reproduction of hydrogel structures of almost any complexity with a spatial resolution of up to 100 µm, providing conditions for preserving the initial physicochemical and biochemical properties of thermolabile substances.

SA scaffold printing was performed in accordance with the following algorithm. The initial composition was loaded under aseptic conditions into a 2 mL dispenser with a nozzle (d = 400 µm). The dispenser and the glass substrate temperatures were maintained at 20 °C and −10 °C, respectively. A simple mesh-like disk with a diameter of 10 mm and a thickness of 3 mm with a mesh size of 1 × 1 mm^2^ was used as a 3D computer model. 

During 3D cryoprinting, hydrogel compositions based on either pure SA (control) or with the addition of genetic constructs were deposited in automatic mode to the glass substrate. After the printing process completion, the fabricated scaffolds were separated from the substrate with a scalpel and transferred in a frozen state in a glass container with 50 mL of 2 wt.% or 10 wt.% CC aqueous solution at 20 °C for 1 h. The samples were then washed three times in diH2O at 20 °C for 5 min. All stages of the described process, starting from the preparation of initial compositions and 3D printing of matrices up to their final washing, were carried out in a sterile laminar box providing aseptic conditions for conducting experiments.

The surface morphology of the SA scaffolds was analyzed with an optical Bresser Advance ICQ stereoscopic microscope (Bresser, Rhede, Germany) equipped with a Levenhuk C510 camera (Levenhuk, Praha, Czech Republic).

For further research, three series of hydrogel scaffolds were manufactured (Figure 9) based on: (1) pure SA (control); (2) SA containing only plasmid DNA (gene-activated with pEGFP); and (3) SA containing polyplexes (gene-activated with pEGFP/TA).

### 5.4. Cell Culture

To analyze biocompatibility and bioresorption of GASs mesenchymal stem cells (MSCs), derived from rat adipose tissue, at 3–4 passages were used. The transfecting ability of polyplexes and the functional characteristics of the released plasmid DNA were analyzed using human embryonic kidney (HEK293) cell line. Cells were incubated in growth medium: DMEM/F12 (PanEco, Moscow, Russia), 10% fetal bovine serum (FBS, PAA laboratories, Etobicoke, ON, Canada), 0.584 mg/mL L-glutamine (PanEco, Russia), 5000 U/mL streptomycin (PanEco, Russia) and 5000 U/mL penicillin (PanEco, Russia) in Petri dishes under standard culture conditions (37 °C, 5% CO_2_).

### 5.5. MTT-Test 

Cytotoxicity of SA scaffolds has been evaluated by a colorimetric method using MTT. MSCs were detached from the surface of Petri dishes using a Versene solution (PanEco, Russia) with the addition of 0.25% trypsin (PanEco, Russia) and seeded in 24-well plates (Corning, NY, USA) with a density of 5 × 10^4^ cells per well. Non-cross-linked SA, control scaffolds cross-linked with 2 wt.% or 10 wt.% CC solutions, as well as gene-activated scaffolds, were placed in Transwell system with a pore size of 8 µm (Corning Transwell). The wells without scaffolds (cells only) were used as controls. After 1 and 7 days, MTT (tetrazolium (3-(4,5-dimethyl thiazolyl−2)−2,5-diphenyltetrazolium bromide, PanEco, Russia) at a concentration of 0.5 mg/mL was added to the wells and incubated for 2 h at 37 °C. Formazan crystals were extracted from cells using dimethylsulfoxide (DMSO, PanEco, Russia) stirring on a shaker for 20 min. The formazan absorption was evaluated by measuring the optical density of the eluate at a wavelength of 570 nm subtracting the background value at 620 nm on a BioRad Reader XMark tablet reader (Bio-Rad Laboratories, Hercules, CA, USA).

### 5.6. Cell Adhesion Study

To study the primary cell adhesion, scaffolds were incubated with MSCs pre-labeled with PKH-26 (Sigma-Aldrich, St. Louis, MI, USA). MSCs were removed from Petri dishes with a Versene solution with 0.25% trypsin and centrifuged at 1100 rpm for 10 min. Cells were stained with PKH-26 dye according to the manufacturer’s protocol. Cells were precipitated by centrifugation after staining and a cell suspension with a concentration of 1 × 10^6^ cells per ml of growth medium was obtained. The matrices were placed in 48-well plates, 25 µL of the cell suspension was added to each well and incubated for 10 min under standard culture conditions, and then 375 mL of growth medium was carefully added. After 5 days, scaffolds with adhered cells were examined using fluorescence Zeiss Axio Observer.D1 microscope (Carl Zeiss Microscopy GmbH, Oberkochen, Germany).

### 5.7. Hydrogel Scaffold Bioresorption In Vitro

To study the SA scaffolds bioresorption process samples were incubated in 24-well plates in a growth medium in the presence of MSCs for 21 days. Structural changes in SA scaffolds, and changes in the fiber diameter, in particular, were evaluated every 7 days using light Zeiss Axio Observer.D1 microscope (Carl Zeiss Microscopy GmbH, Oberkochen, Germany). The swelling degree of SA fibers α was calculated using the equation:(1)$$\alpha =\frac{d-d_{0}}{d_{0}}*100\%,$$
where *d*_0_ and *d* are initial and current fiber diameters, correspondently.

### 5.8. Structural Stability of the Genetic Constructs in Hydrogel Scaffolds

To study the influence of the 3D cryoprinting on the pDNA molecular structure preservation a spectrophotometric analysis of plasmid released from an SA scaffold into a saline solution (NanoPhotometer P330, Implen, Munich, Germany) was assessed. The spectrum of the released pDNA was compared with the characteristic spectrum measured for the pEGFP with the concentration of 10 µg/mL (Initial pEGFP).

The transfecting ability of the released plasmids was evaluated on HEK293 cell culture. HEK293 were detached from the surface of Petri dishes using a Versene solution with 0.25% trypsin and seeded into 24-well plates with a density of 1 × 10^5^ cells per well. Plasmids released from gene-activated scaffolds with the addition of PEI in a ratio of 1:3 were added to the wells. The wells without plasmid DNA (cells only) were used as controls. Transfection was evaluated after 24 h.

### 5.9. In Vivo Study

All experiments were approved by the local bioethical committee of Sechenov University (No. PRC-079 from 6 April 2021) in compliance with the Guide for the Care and Use of Laboratory Animals published by the US National Institutes of Health (NIH publication no. 85–23, revised 1996), European Convention for the Protection of Vertebrate Animals used for Experimental and Other Scientific Purposes, and ISO 10993–22006.

The tissue response to the hydrogel scaffolds was evaluated using intramuscular implantation in male Wistar rats (n = 9) weighing 200 g. Laboratory animals were anesthetized with intramuscular injection of 30 mg/kg Zoletil (Virbac, Carros, France) and 5 mg/kg Xylazine (Interchemie Werken “de Adelaar” BV, Venray, The Netherlands). After shaving and disinfection a longitudinal incision of the skin in the posterior thigh was made. Scaffolds were implanted inter muscle of posterior thigh 1 cm deep. The wound was sutured with Vicryl 5/0 (Ethicon, Guaynabo, PR, USA). To prevent infectious complications, 10 µg/kg of Ceftriaxone (Biochemist, Moscow, Russia) was intramuscularly injected. 

14 days after surgery all rats were euthanized by CO_2_ inhalation, the implantation sites were resected and fixed in 10% (Labiko, Saint-Petersburg, Russia) formalin for 24–48 h.

### 5.10. Histological and Immunohistochemical Assays 

Formalin-fixed samples of implanted SA scaffolds were embedded in paraffin and cut into 5 µm-thick sections (HistoCore Arcadia C, Leica, Wetzlar, Germany) according to the standard procedure [39]. The sections were then stained with hematoxylin and eosin (HE). The degree of inflammation was assessed by counting the number of inflammatory cells in several regions of interest (ROIs) (at least 5 for each sample). 

For the immunohistochemical (IHC) detection of pEGFP transfected cells, antigen retrieval was performed in a citrate buffer (pH = 6.0) for 30 min at 98 °C. Then the sections were incubated with rabbit polyclonal anti-EGFP antibody (1:100; Evrogen, Moscow, Russia), followed by incubation with goat anti-rabbit IgG antibodies conjugated with peroxidase (1:50; Imtek, Moscow, Russia). 3,3′-diaminobenzidine tetrahydrochloride (DAB, Merck KGaA, Darmstadt, Germany ) was used as chromogen. To detect cells, the sections were stained with hematoxylin and examined using light microscopy (Zeiss Axio Observer.D1, Carl Zeiss Microscopy GmbH, Oberkochen, Germany). For a comparative assessment of the transfection efficacy, the “transfection depth” was determined as the distance deep into the tissue from the scaffold surface, on which cells synthesizing the EGFP protein can still be detected, and the number of transfected cells in several ROIs was calculated (at least 5 for each sample).

### 5.11. Statistical Analysis

Statistical analysis and graphing were performed with SigmaPlot v14.0 (Systat Software Inc., Palo Alto, Santa Clara, CA, USA). The differences between groups were assessed by one-way ANOVA using Tukey post hoc tests. Statistical significance was accepted for *p* < 0.05.

## Figures and Tables

**Figure 1 gels-08-00421-f001:**
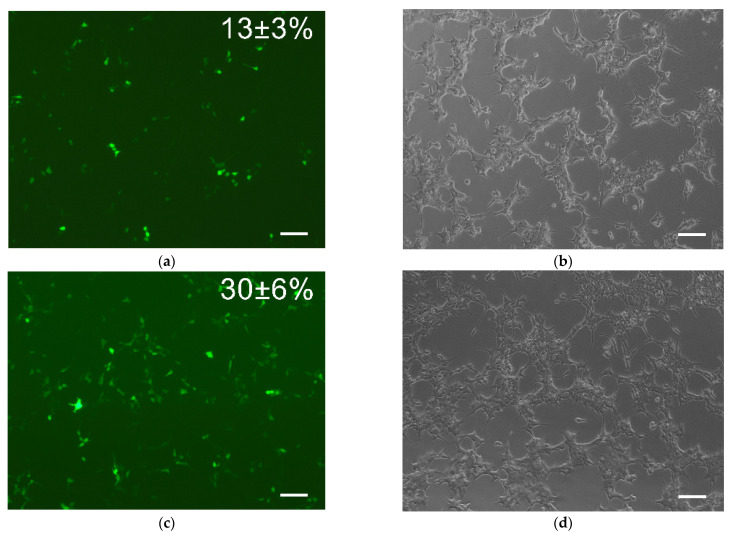
Transfection efficiency. (**a**–**d**) Transfected HEK293 after 24 h incubation with (**a**) 1 µg pEGFP:2 µL TF and (**b**) 1 µg pEGFP:3 µg PEI polyplexes; (**a**,**b**) Fluorescent and (**c**,**d**) light microscopy. Scale bar 100 µm.

**Figure 2 gels-08-00421-f002:**
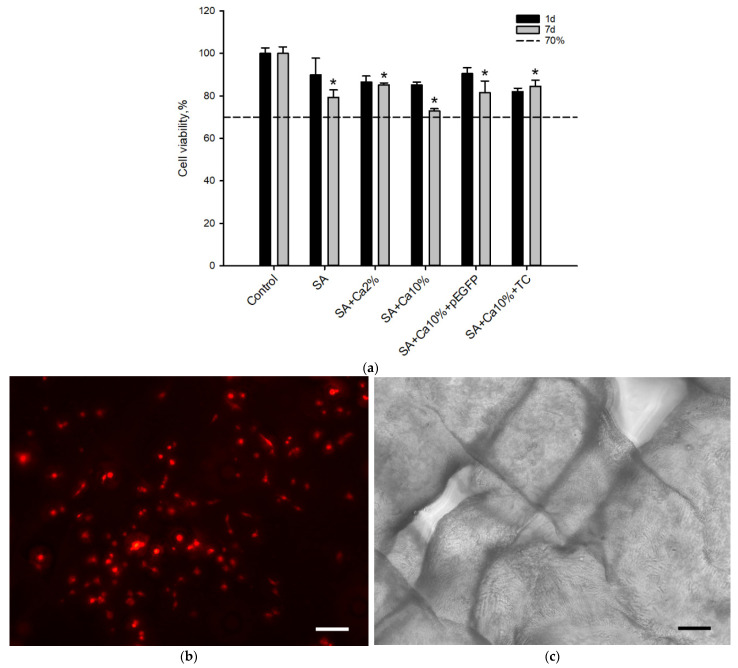
In vitro biocompatibility. (**a**) SA scaffolds cytotoxicity assessment for 1 and 7 days. MTT test. The optical density of the control group was taken as 100%. *—*p* < 0.05 (relative to the initial sodium alginate); (**b**,**c**) MSC adhesion on the SA scaffold surface. PKH−26. (**b**) Fluorescent and (**c**) light microscopy. Scale bar 100 µm.

**Figure 3 gels-08-00421-f003:**
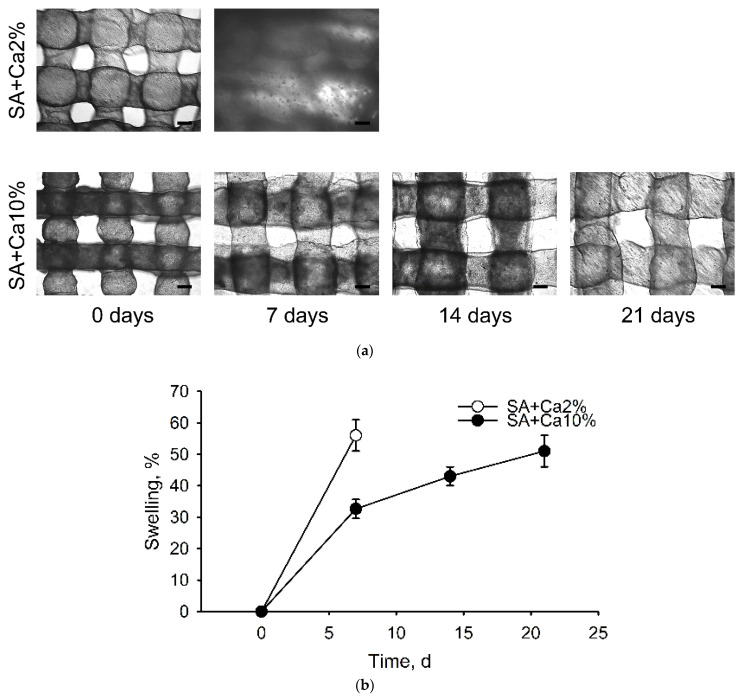
Alginate scaffold swelling. (**a**) The structure of SA scaffold fibers after 0, 7, 14, and 21 days of their incubation with MSCs. Light microscopy. Scale bar 200 µm; (**b**) The swelling degree of SA matrices crosslinked with 2 wt.% and 10 wt.% CC solutions over time.

**Figure 4 gels-08-00421-f004:**
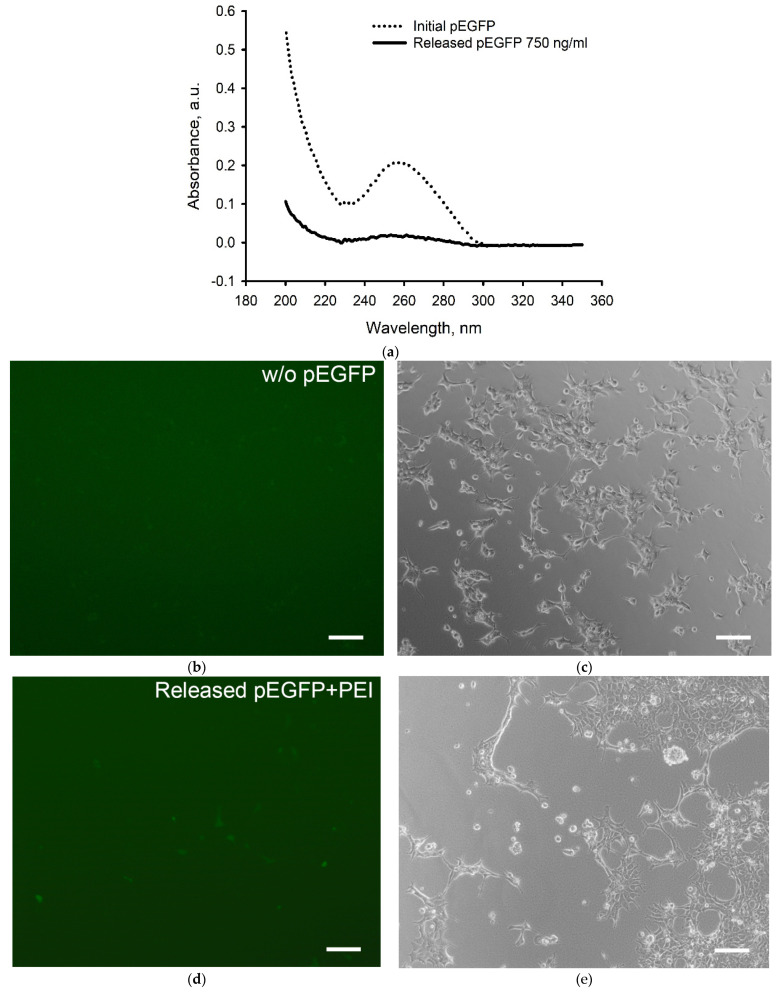
pEGFP stability assessment. (**a**) UV spectrum of pEGFP released from the alginate scaffold during 7 days of its incubation in saline solution. UV spectrophotometry; (**b**–**e**) Transfection of HEK293 incubated (**b**,**c**) without and (**d**,**e**) with addition of pEGFP:PEI (1:3), after 24 h; (**b**,**d**) Fluorescent and (**c**,**e**) light microscopy. Scale bar 100 µm.

**Figure 5 gels-08-00421-f005:**
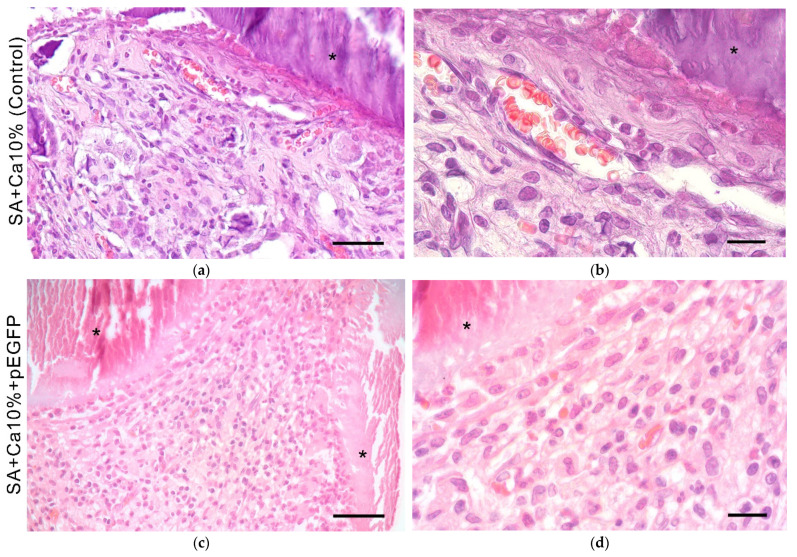
In vivo biocompatibility. (**a**–**f**) Implantation of SA-based GAS onto the muscle tissue in rats. 14 days. HE staining. Light microscopy. *—implanted scaffold. Scale bar (**a**,**c**,**e**) 50 µm and (**b**,**d**,**f**) 100 µm; (**g**) Quantitative assessment of the inflammation degree during the sodium alginate-based scaffolds implantation. *—*p* < 0.05 (relative to control-SA + Ca10%).

**Figure 6 gels-08-00421-f006:**
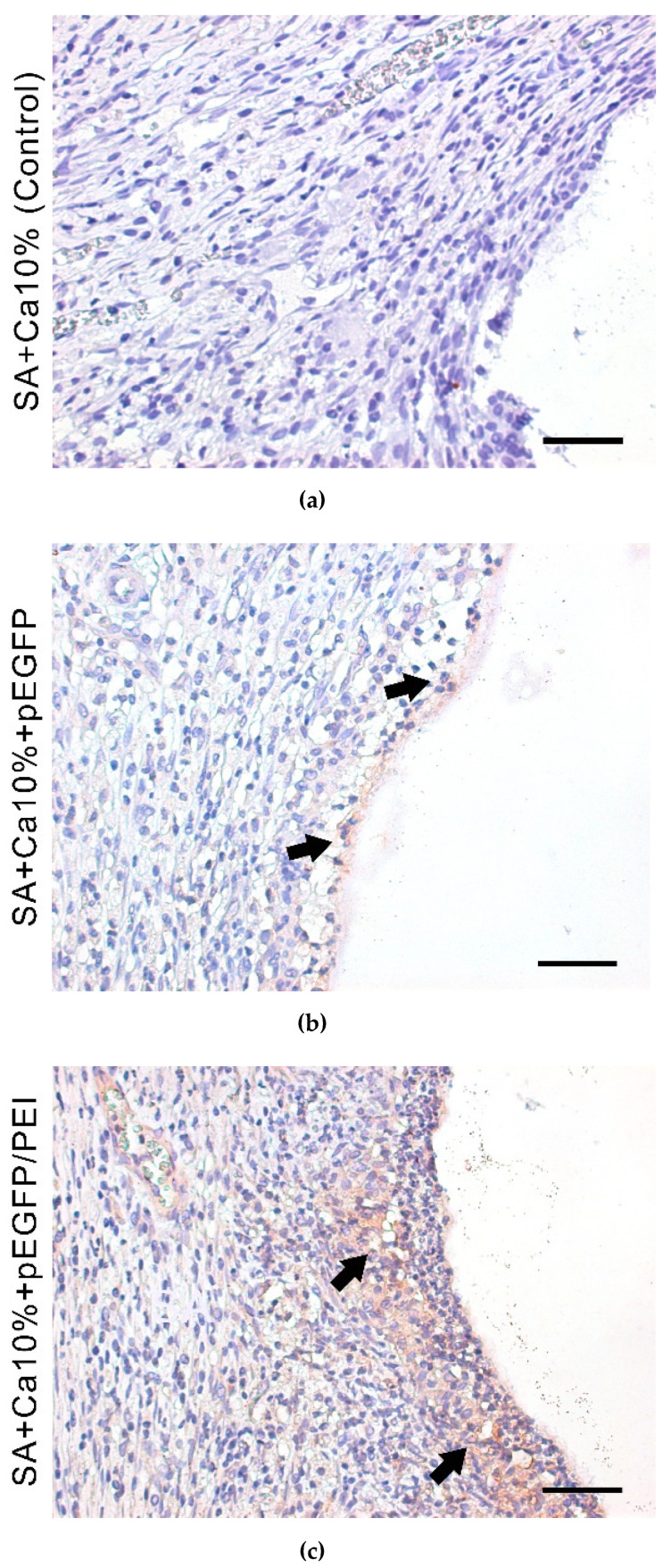
Transfection in vivo with SA scaffolds without pEGFP (**a**), gene-activated with pEGFP (**b**) and gene-activated with pEGFP/PEI (**c**). Immunohistochemical detection of EGFP-producing cells. 14 days after implantation. Chromogen–DAB. The nuclei are stained with hematoxylin. Light microscopy. Scale bar 50 µm.

**Figure 7 gels-08-00421-f007:**
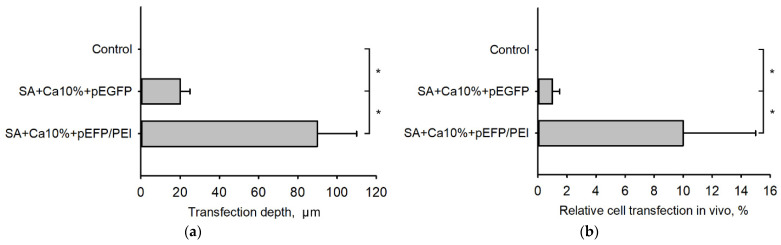
Quantitative assessment of the transfection in vivo; (**a**) The transfection depth in the tissue; (**b**) Number of transfected cells in the SA scaffold implantation zone. *—*p* < 0.05 (vs. control (SA + Ca10%)).

**Figure 8 gels-08-00421-f008:**
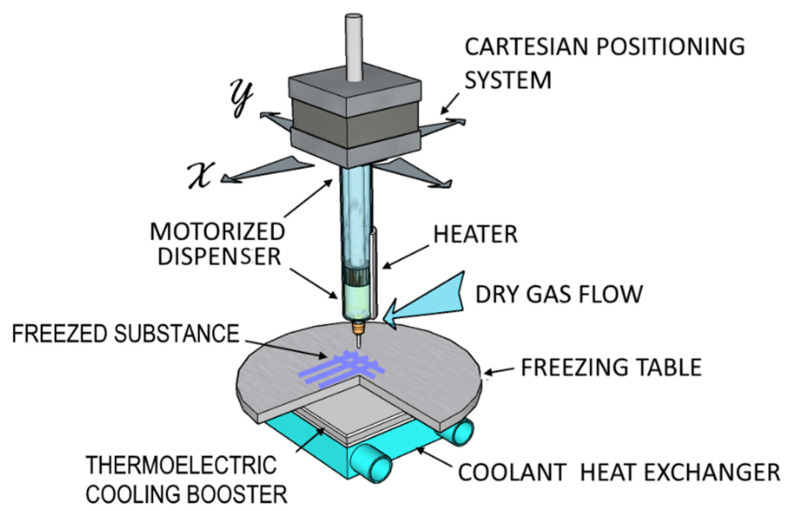
Schematic representation of 3D cryoprinter.

**Figure 9 gels-08-00421-f009:**
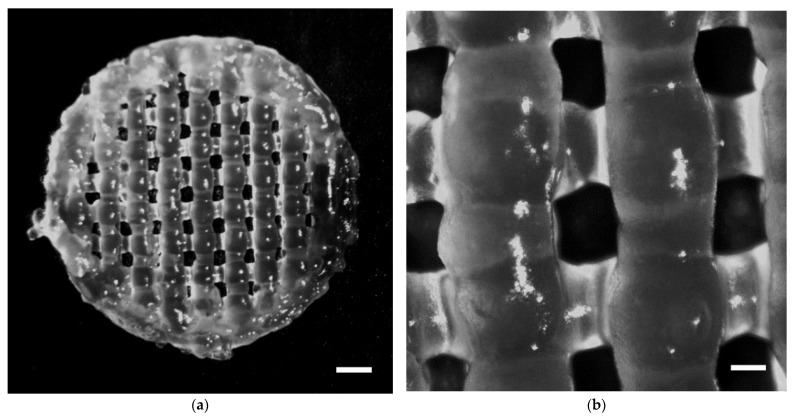
Light microscopy images of sodium alginate hydrogel scaffold. Scale bar (**a**) 1 mm and (**b**) 200 µm.

**Table 1 gels-08-00421-t001:** Size distribution and zeta-potentials of plasmid DNA, TAs, and polyplexes. DLS.

	Hydrodynamic Diameter, nm	Zeta-Potential, mV
pEGFP	420 ± 70	−17 ± 6
TF	700 ± 100	+3.0 ± 0.2
PEI	310 ± 110	+6.0 ± 1.0
pEGFP/TF	140 ± 20	+1.1 ± 0.5
pEGFP/PEI	90 ± 20	+23 ± 8

## Data Availability

Available upon request.

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
