# Peer review of "3D Printed Gene-Activated Sodium Alginate Hydrogel Scaffolds"

_gels, 2022, doi:10.3390/gels8070421_

Round 1
Reviewer 1 Report
The manuscript by Khvorostina et al describes the use of sodium alginate hydrogels as gene (or polyplex) delivery agents. The text is very concise and sometimes lacks key information for the reader to fully understand what the authors did perfom. The figure legends are kept to a bare minimum which imposes the reader to continously look at the M & M section to understand the full significance of the panels. This is definitely not comfortable.
P2, Results: The size distribution of the polyplexes is multimodal, both in the case of TF and PEI (fig 1). This is not so usual and deserves special comment. Do the smallest nanoparticles correspond to polyplexes, and the larger ones to excess TA? What about the zeta potential value for these two populations of particles? Analyses should be improved.
P5, fig 4a: What does "Initial pEGFP 10 mkg/ml" stand for? Please check. What the authors try to show is not clear. They should improve the figure legend to let the reader understand the experiment they carried out.
P7, lines 181-185: The comments refer to figure 7, and not to figure 9 as indicated.
P8, lines 194-198: Please remove these sentences as they are already included in the previous paragraph.
P9: The legend of figure 7 does not refer to the displayed data. Do statistics refer to control? Please state it clearly. What does mean "mkm" on the x axis? An additional control, pEGFP/PEI, would bring valuable information about efficiency of the SA hydrogel to improve transfection. It is definitely missing and should be included and discussed in a revised version.
P11, line 292: "8 mg of sodium alginate (Chimmed, Russia) was dissolved in 92 ml of water to obtain 8 wt. % solution". This procedure does not furnish a 8% solution. 8 mg or 8 g of sodium alginate, 92 ml or 92 µL of water? Please check.
Lines 357, 392…: Please indicate correctly the power of ten used.
Author Response
Reviewer Report #1.
- P2 (original manuscript). Results: The modality of size distribution can be explained by the aggregation of polyplexes [1,2] resulting in the appearance of the second peak with higher hydrodynamic diameter. The smallest nanoparticles correspond to the disperse polyplexes and provide sufficient information about the real polyplex size and pDNA condensation with transfecting agents. We have also conducted extra experiment and added zeta potential values for polyplexes with different transfecting agents. Considering the mentioned above and the fact that the aim of the experiment was to determine the more effective TA, we have decided to present this data using Table 1.
[1] E. Lai and J. H. Van Zanten, “Monitoring DNA/poly-L-lysine polyplex formation with time-resolved multiangle laser light scattering,” Biophys. J., vol. 80, no. 2, pp. 864–873, 2001, doi: 10.1016/S0006-3495(01)76065-1.
[2] Y. Wu, Z. Fei, L. J. Lee, and B. E. Wyslouzil, “Coaxial electrohydrodynamic spraying of plasmid DNA/polyethylenimine (PEI) polyplexes for enhanced nonviral gene delivery,” Biotechnol. Bioeng., vol. 105, no. 4, pp. 834–841, 2010, doi: 10.1002/bit.22583.
- P5. Fig 4a (original manuscript): was corrected according to the reviewer’s comment in Materials and Methods section.
- P7, lines 181-185 (original manuscript): were corrected according to the reviewer’s comment.
- P8, lines 194-198 (original manuscript): were corrected according to the reviewer’s comment.
- P9 (original manuscript): The legend of figure 7 as well as statistics were corrected according to the reviewer’s comment. “mkm” were changed to “µm” in figure 7.
The application of biopolymer scaffold for gene delivery is the effective way to overcome the drawbacks of the direct gene delivery: turnover of polyplexes in bloodstream, unpredictable gene localization and inflammatory reaction caused by the high doze. Our in vivo experiment was well-considered and carefully designed according to the worldwide scientific practices [3,4]. Moreover, polyplexes are not usually used for direct gene delivery in contrast to adenoviruses and physical methods [5]. Thus, in our opinion, the inclusion of the additional control, pEGFP/PEI, in the in vivo experiment is unnecessary.
[3] J. Bonadio, E. Smiley, P. Patil, and S. Goldstein, “Localized, direct plasmid gene delivery in vivo: Prolonged therapy results in reproducible tissue regeneration,” Nat. Med., vol. 5, no. 7, pp. 753–759, 1999, doi: 10.1038/10473.
[4] R. Yang, F. Chen, J. Guo, D. Zhou, and S. Luan, “Recent advances in polymeric biomaterials-based gene delivery for cartilage repair,” Bioact. Mater., vol. 5, no. 4, pp. 990–1003, 2020, doi: 10.1016/j.bioactmat.2020.06.004.
[5] G. Pelled et al., “Direct gene therapy for bone regeneration: Gene gelivery, animal models, and outcome measures,” Tissue Eng. - Part B Rev., vol. 16, no. 1, pp. 13–20, 2010, doi: 10.1089/ten.teb.2009.0156.
- P11, line 292 (original manuscript): were corrected according to the reviewer’s comment.
- Lines 357, 392 (original manuscript): were corrected according to the reviewer’s comment.

Reviewer 2 Report
Khvorostina et al presents an interesting study using alginate hydrogel for the release of DNA nanocomplexes for in vitro and in vivo gene transfection. Gene therapy is rapidly rising recently and there is still strong needs for methods to deliver plasmid nanocomplexes. Therefore, the current paper addresses an important question with new data on alginate-based 3D printed systems. I recommend the acceptance of the paper after the following changes:
1. Are there duplicates in experiments shown in Fig 1, 5, 6? I recommend the addition of graph showing the quantification of these data.
2. What’s the release profile for the DNA from the 3D printed alginate hydrogel?
Author Response
Reviewer Report #2.
- There were at least 5 duplicates for in vitro and 3 for in vivo experiments shown in Fig 1, 5, 6. Standard errors were added to the Fig 1. Graphs 5g and 7a,b show the quantification of data presented in Fig 5 and 6, correspondently.
- While preparing the paper spectrophotometric analysis of plasmid released from 3D printed hydrogel scaffolds into saline solution (NanoPhotometer P330, Implen, Germany) was assessed (Fig. 1). The investigation demonstrated gradual release of pDNA in saline solution for 4 months before complete scaffold degradation. Since the degradation processes in vitro and in vivo are usually different due to the complexity of in vivo model, we decided to eliminate these results from the original paper so as not to confuse the reader and leave it for further research.

Round 2
Reviewer 1 Report
As described by the authors, measurement of zeta potentials is not appropriate. The M&M section states that they were measured in deionized water. What was the value of pH? As zeta potential of titratable species is highly dependent on pH, they should have been measured at physiological pH (ca. 7.4) for relevant discussion. Besides, "swelling" can occur along with ionization of nanoparticles. Consequently, significance of the hydrodynamic diameter given under the conditions used by the authors may also be flawed.
P2 (original manuscript). Results: The modality of size distribution can be explained by the aggregation of polyplexes [1,2] resulting in the appearance of the second peak with higher hydrodynamic diameter. The smallest nanoparticles correspond to the disperse polyplexes and provide sufficient information about the real polyplex size and pDNA condensation with transfecting agents. We have also conducted extra experiment and added zeta potential values for polyplexes with different transfecting agents. Considering the mentioned above and the fact that the aim of the experiment was to determine the more effective TA, we have decided to present this data using Table 1.
According to the authors's interpretation, the larger particles woul correspond to aggregates of polyplexes. However, these aggregates are similar in size to "unloaded" carriers, either TF (700 + 100 nm) or PEI (310 ± 110 nm). No significant difference between the "unloaded" carriers and those putative aggregates can be appreciated, so the authors' interpretation is not supported by the data. Possibly, a more relevant discussion would arise from measurements under controlled pH values.
Author Response
Dear Assistant Editor: Mr. Saran Bannatong,
On behalf of my co-authors, I would like to thank you for the opportunity to revise and resubmit our manuscript gels-1766523, entitled “3D Printed Gene-activated Sodium Alginate Hydrogel Scaffolds”. It is important mentioning that we found the discussion with the reviewers to be valuable for our manuscript and understanding of further studies directions.
- The usage of deionized water in measuring of hydrodynamic diameter and zeta-potential is justified by the physiological pH value (pH 7.0 was added in the text) and the application of water as polyplex solvent by other scientific groups [1], [2]. pH and ions indeed have influence on size and charge of nanoparticles [3], but there are also studies showing that characteristic polyplex parameters values are more stable in water [4]. Thus, we consider the values provided to be appropriate for publishing.
[1] H. Zhang, Z. Chen, M. Du, Y. Li, and Y. Chen, “Enhanced gene transfection efficiency by low-dose 25 kDa polyethylenimine by the assistance of 1.8 kDa polyethylenimine,” Drug Deliv., vol. 25, no. 1, pp. 1740–1745, 2018, doi: 10.1080/10717544.2018.1510065.
[2] K. Atluri, D. Seabold, L. Hong, S. Elangovan, and A. K. Salem, “Nanoplex-Mediated Co-delivery of Fibroblast Growth Factor and Bone Morphogenetic Protein Genes Promotes Osteogenesis in Human Adipocyte-Derived Mesenchymal Stem Cells,” Mol Pharm., vol. 12, no. 8, pp. 3032–3042, 2015, doi: 10.1021/acs.molpharmaceut.5b00297.Nanoplex-Mediated.
[3] N. Bono, F. Ponti, D. Mantovani, and G. Candiani, “Non-viral in vitro gene delivery: It is now time to set the bar!,” Pharmaceutics, vol. 12, no. 2, 2020, doi: 10.3390/pharmaceutics12020183.
[4] I. González-Domínguez, E. Puente-Massaguer, J. Lavado-García, L. Cervera, and F. Gòdia, “Micrometric DNA/PEI polyplexes correlate with higher transient gene expression yields in HEK 293 cells,” N. Biotechnol., vol. 68, pp. 87–96, 2022, doi: 10.1016/j.nbt.2022.02.002.
- We agree that the nature of the modality of nanoparticles size distribution is unclear and requires further investigation. The aim of our study was to compare the complexation efficacy of two transfecting agents under same conditions, which applicability we hope to be clarified in p1. The investigation of the processes accompanying polyplex formation as well as different medium and pH influence on the size distribution and zeta-potential could be the main challenge in our next study.
Corresponding changes are tracked in the manuscript text in the revised file.
Thank you again for your consideration of our revised manuscript.
Best regards,
Prof., Dr. Sci., PhD
Vladimir S. Komlev
